# Investigation of the neural correlation with task performance and its effect on cognitive load level classification

**Farzana Khanam** [1,2]*, **Mohiuddin Ahmad** [3], **A. B. M. Aowlad Hossain** [4]

1 Department of Biomedical Engineering, Khulna University of Engineering & Technology (KUET), Khulna, Bangladesh, 2 Department of Biomedical Engineering, Jashore University of Science and Technology (JUST), Jashore, Bangladesh, 3 Department of Electrical and Electronic Engineering, Khulna University of Engineering & Technology (KUET), Khulna, Bangladesh, 4 Department of Electronics and Communication Engineering, Khulna University of Engineering & Technology (KUET), Khulna, Bangladesh

☯ These authors contributed equally to this work.
* farzanabme@just.edu.bd

**Data Availability Statement:** Raw Data: This research work was conducted on third party data which is publicly available in: https://physionet.org/physiobank/database/eegmat/. All rights of the data are reserved by Institute of Biology and Medicine of

## Abstract

Electroencephalogram (EEG)-based cognitive load assessment is now an important assignment in psychological research. This type of research work is conducted by providing some mental task to the participants and their responses are counted through their EEG signal. In general assumption, it is considered that during different tasks, the cognitive workload is increased. This paper has investigated this specific idea and showed that the conventional hypothesis is not correct always. This paper showed that cognitive load can be varied according to the performance of the participants. In this paper, EEG data of 36 participants are taken against their resting and task (mental arithmetic) conditions. The features of the signal were extracted using the empirical mode decomposition (EMD) method and classified using the support vector machine (SVM) model. Based on the classification accuracy, some hypotheses are built upon the impact of subjects' performance on cognitive load. Based on some statistical consideration and graphical justification, it has been shown how the hypotheses are valid. This result will help to construct the machine learning-based model in predicting the cognitive load assessment more appropriately in a subject-independent approach.

## Introduction

Beginning the mid of the twentieth century, researchers moved towards cognitive science in simultaneous nature through the research advancement about the human mind or psychology [1]. Cognitive science endorses detailed instrumental investigation of the human brain to illuminate the functional connectivity of the different mental states. Cognitive science yields the term 'Cognition' to acquire and evaluate knowledge through experience and the senses, which are accountable for mental actions or processes [2]. In the late 1980s, a study of solving problems promoted modern cognitive load theory by John Sweller [3]. Cognitive load theory hypothesizes that the human brain holds a limited capacity for working memory termed

Taras Shevchenko National University of Kyiv. So, the authors don't need to share this data further. Statistical Data: All numerical findings of this work are shared in the manuscript in detail. The numerical data are shown either in Table or Graph. The data (result) of Fig 6, Fig 7, Fig 8, and Fig 9 are given in supplementary materials. Code: The code of topoplot presentations (for Fig 10, Fig 11, and Fig 12) is modified from the code given in: https://www.mathworks.com/matlabcentral/fileexchange/69991-topoplot-for-b-alert-x-10-9-channel-eeg-signal Easily, anyone can recreate and customize the code available here.

**Funding:** The author(s) received no specific funding for this work.

**Competing interests:** The authors have declared that no competing interests exist.

cognitive load and if existing information exceeds this capacity, information overload occurs [4]. On account of the cognitive load theory, several scientists have researched on human brain's working capacity to reflect the scenario of reducing over-cognitive load [5]. This cognitive load measurement is an interesting research area to understand the process of building working memory, learning procedures, human cognitive architecture, etc. Brain functionality generates due to some cognitive work and that's why this functionality is correlated with the cognitive loads of the human brain. There are several non-invasive techniques to observe brain functionality like functional magnetic resonance imaging (fMRI), functional near-infrared spectroscopy (fNIRS), magneto-encephalogram (MEG), EEG, etc. [6]. Both the fMRI and MEG are very sophisticated but due to the cost, weight, and high noise sensitivity, these modalities have very limited popularity in brain- computer interface (BCI) applications. On the other hand, in the last two decades, fNIRS got popularity in cognitive workload and BCI-related research [7–10]. However, this modality has temporal resolution issues. Due to being cheap, portable, and having a high temporal resolution, EEG is a very popular modality to observe brain functionality. Over a long period, EEG is widely used in the field of cognitive load assessment [11–13].

There are several types of research works on human EEG-based cognitive load analysis such as mental arithmetic [14], *n*-back tasks [15], word count [16, 17], video games [18, 19], tasks in a simulated environment [20, 21], etc. Cognitive load studies reflect the scenario of how hard the brain is working to solve tasks. Most of the articles have classified different cognitive load levels with the help of machine learning-based classification algorithms. In [22], EEG signals are captured in terms of three types of taste responses of autistic children to define different cognitive load levels. The research achieved the classification results of 92.3% among three classes of autism severity (mild, moderate, and severe) with the help of a multilayer perceptron neural network. The researchers in [23] acquired three types of EEG data from a relaxed position, a reading task, and a mathematical calculation task from healthy participants. Several machine learning classifiers were used to classify the mentioned three classes of cognitive loads, whereas the logistic linear classifier showed the highest accuracy of 92%. Moreover, a recent work [24] portrayed the classification of three classes of cognitive load levels (Low-Medium-High) through *n*-back tasks (*0*-back, *2*-back, and *3*-back). This work achieved a classification accuracy of 77.20% and 87.89% for 3-class and 2-class, respectively using the SVM classifier.

From the literature review, it is observed that most of the research works concluded their results in a subject-dependent approach. The article [25] classified the cognitive load levels as low (*1*-back), medium (*2*-back), or high (*3*-back) through *n*-back tasks. The EEG data were classified by three deep learning subject-specific classifiers: convolutional neural network (CNN), long short-term memory network (LSTM), and convolutional LSTM network (Conv-LSTM) and reached the best accuracy of 97.92% from feature combination of phase locking value with CNN and Conv-LSTM. Recent work [26] utilized the SVM model to classify the two classes (high-low) of cognitive workload (*0*-back vs *2*-back & *3*-back) with the highest accuracy of 94.44% and an average accuracy of 75.85% for a 50:50 train test ratio.

However, we hardly found EEG-based cognitive load analysis research works on the basis of subject-independent approaches. The researchers in [27] recognized EEG signals between mental arithmetic and silent reading tasks accomplishing both subject-dependent and subject-independent analysis through the k-Nearest Neighbor algorithm. Considering the abovementioned research, the main reason to survey subject-independent cognitive load study is to establish the generalized approach of performance-based cognitive load level estimation. In the subject-independent approach, the classification accuracy is usually decreased [28]. This may happen for the variation of subjects' performance and experimental situation.

Nevertheless, as far as the knowledge of the authors; there is no research work on cognitive load analysis that considered the subject's experience related to experimental cognitive tasks. This overlooked issue may arise the question "Does a task create the same cognitive workload for all participants irrespective of their expertise or prior knowledge?" Simply, the answer is 'no'.

For instance, suppose that a mathematical problem or $n$-back task (reflects as cognitive task) may create a different amount of cognitive load for different subjects. The complexity levels of the tasks may vary from person to person. A simple arithmetic task can be found difficult for a subject who is weak in mathematics. On the other hand, a man who has a very good knowledge of mathematics can solve even a complex arithmetic task without getting any mental load. Therefore, it is clear that any arithmetic or memorizing task can create different levels of cognitive load for the subjects based on their experiences in that task. It doesn't need to happen in the same way as we are thinking, though these facts motivate us to develop a hypothesis to find out the facts about the subjects' neural correlation with their performances. To the best of the author's knowledge, this type of hypothesis has not been discussed earlier. This is why almost all datasets are constructed with the task and their corresponding EEG signals. Mostly available EEG datasets related to cognitive tasks didn't count the participant's task performance. Therefore, to prove our exclusive hypothesis; only one dataset we got in our hands. The whole methodological validation of the proposed hypothesis is investigated in this paper considering the dataset available in [29, 30].

In this paper, cognitive states (rest and task) from EEG signals are classified considering a subject-independent approach. To train the classifier model, the features of the EEG signals were extracted using the empirical mode decomposition (EMD) technique. Since EEG poses non-stationary property, EMD is a proven feature extraction method to achieve the best classification accuracy. Additionally, it is a data-adaptive multi-resolution technique. Apart from that, the selection of classification algorithms is also important considering some facts such as stability, a trade-off with performance, computational cost, etc. Following our objectives, a developed support vector machine (SVM) model is chosen for simple computational operation and risk minimization [31]. Consequently, the effect of the subject's performance on machine learning-based model training and classification accuracy is extensively investigated. Besides, it is also statistically shown how good and bad performances affect the cognitive load and its classification accuracy acquired from the machine learning model. Some hypotheses are made and proved based on the output of classification accuracy and graphical representations. Finally, to solve the issue, a special proposal is recommended and also evidenced that this proposal can solve the conventional shortcomings and achieve acceptable classification accuracy in subject-independent cognitive load level classification.

## Materials and methods

### Materials

**Dataset description.** In this research, a recognized database has been selected for the collection of EEG signals during mental task activity. The benchmark dataset is available in an open-access repository 'PhysioNet' through the 'PhysioBank' platform, popular for numerous types of medical research data [29, 30]. EEG recordings and corresponding reference background is preserved in the dataset according to the international 10/20 system and organized conferring to BIDS standard [31, 32]. The dataset was chosen to investigate EEG signals correlating cognitive activity during an intensive mental arithmetic task e.g., serial subtraction [33]. The experimental protocol was approved by the Bioethics Commission of the Educational and Scientific Centre at the "Institute of Biology and Medicine", at the Taras Shevchenko National

University of Kyiv. Each participant was prior informed about the experimental protocol and signed in written consent conferred to the Helsinki Declaration [34].

**Participants.** Initially, 66 (sixty-six) healthy right-handed students participated in the experiment following prior specified instructions. Among the participants, 47 were women and 19 were men, and the age ranges were from 18 to 26 years (18.25 ±2.17) including normal visual perception and color vision abilities. They had no clinical assertions of mental disorders or learning (verbal and non-verbal) disabilities. The participants were free from any addiction and psychological or neurological complaints. Through EEG visual examination, 30 (thirty) participants' data were disqualified from the database owing to poor EEG quality and excluded electrooculography (EOG) and electromyography (EMG) artifacts. Consequently, the ultimate sample size was finalized for 36 (thirty-six) participants (27 women and 9 men). From several research studies, it is found that performing different arithmetic tasks generate psycho-physiological stress which can be considered for cognitive workload experimental protocol [35–37]. So, an arithmetic task e.g., serial subtraction of two numbers is chosen as a mental workload in this research protocol.

**Experimental protocol.** At first, the participants were told to sit in an armchair comfortably in a dark sealed chamber ambiance. The whole experimental protocol was divided into three parts: a) pre-resting state; b) resting state; and c) task state. While pre-resting state, the participants were asked to follow the instructions about the arithmetic task and resting state. Then, the participants were told to count mentally in a silent manner without any movements in the precise tempo. Basically, these activities were performed for 3 minutes for being compliant with the whole experimental situation. After that sequence, the resting state began and the participants kept their eyes closed to feel relaxed for another 3 minutes. Then the mental arithmetic task (serial subtraction) was performed of 4-digit minuend and 2-digit subtrahend numbers (e.g., 6543 and 12, 3472 and 47, etc.). During the task, each trial started with oral communication. This process was accomplished to intend for 4 minutes. After finishing the task period, the participant had to report all the subtraction results. But in the EEG recordings database, the last 3 minutes of the resting period (excluding the first 3 minutes of experimental or resting adaption) and the first minute (out of 4 minutes) of the mental arithmetic task were composed and stored for further research. These database periods were chosen, because of the simultaneous mental task execution of participants along with varying psychological states due to task performance and overload cognitive activity. The whole aforementioned experimental procedure is portrayed briefly in Fig 1.

To check the result accuracy reported by the participants, a comparable approach was applied to evaluate the quality of task performance. A similar type of comparable task quality evaluation study had been explored in [38]. A mental arithmetic score was estimated based on the comparative correct results for each participant. If the participants' reported result was around 20% of the correct value, then it is considered a successful engagement of that participant.

The individual task performance was analyzed based on the number of arithmetic subtraction operations per minute, which divided the participants into two groups (BAD and GOOD). Group BAD of 10 participants, (mean no. of operations = 6.2 ± 3.37), the task was found difficult. On the other hand, the task was managed without difficulty in group GOOD of 26 participants, (mean no. of operations = 21.99 ± 7.46). The complete research protocol for this work is shown in the following flowchart of Fig 2.

**Data acquisition.** During the data acquisition process, the EEG signals were recorded through Neurocom monopolar EEG 23-channel system (XAI-MEDICA, Ukraine product). Following the internationally recognized nomenclature 10/20 system, the Ag/AgCl electrodes were spatially placed on the scalp [32]. 19 (Nineteen) electrodes were positioned in the

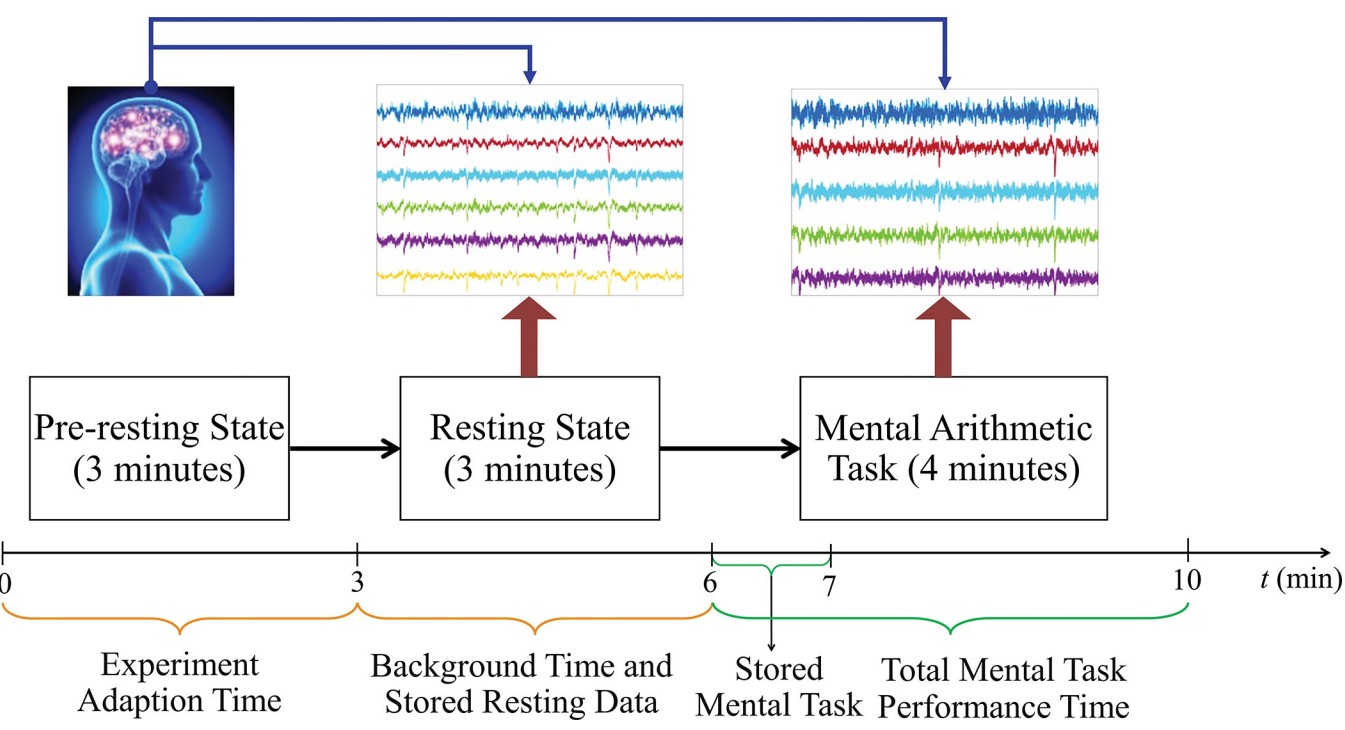

**Fig 1. The brief depiction of performed activity schedule and storage time span during EEG recordings.**

following recording sites: *i*) symmetrical anterior frontal (Fp1, Fp2), *ii*) frontal (F3, F4, Fz, F7, F8), *iii*) central (C3, C4, Cz), *iv*) parietal (P3, P4, Pz), *v*) occipital (O1, O2), and *vi*) temporal (T3, T4, T5, T6) and symbolized nineteen (19) channels, correspondingly. The other two ear electrodes (A1, A2) were used as reference electrodes (20th channel (single)). Subsequently, the 21st channel recorded ECG signals, and the 22nd channel contained timing in the European data format (EDF) and characteristics of the stimulus. In this dataset, channel number 23 is not recognized. However, the ECG signal contents are not related to this study. The whole electrode placement is depicted in Fig 3. The impedance value of below 5kW was maintained among interconnected electrodes.

## Methods

**Pre-processing.** The EEG signals were already pre-processed. For filtering, a 0.5Hz cut-off frequency high-pass filter and a low-pass filter with 45 Hz were applied. Apart from that, a 50 Hz power line noise (notch) filter was also applied. The sampling rate was 500 Hz for each channel. Therefore, the dataset is directly employed for the research purpose. The EEG data files were provided in EDF format. The recording files are comprised of separate artifact-free EEG segments.

**Methodology.** From various research works, it is known that EEG signals are non-stationary and highly random. Time-domain-based feature extraction is not enough to acquire appropriate information. Moreover, frequency domain features often fail to detect the time-wise frequency variation [24]. Considering these actualities, EEG signal features should be extracted to obtain instantaneous phase and frequency information for time-varying data analysis. In this research work, EMD is employed to decompose EEG signal features. EMD generates a highly localized time-frequency estimation of a signal in the modulated oscillatory

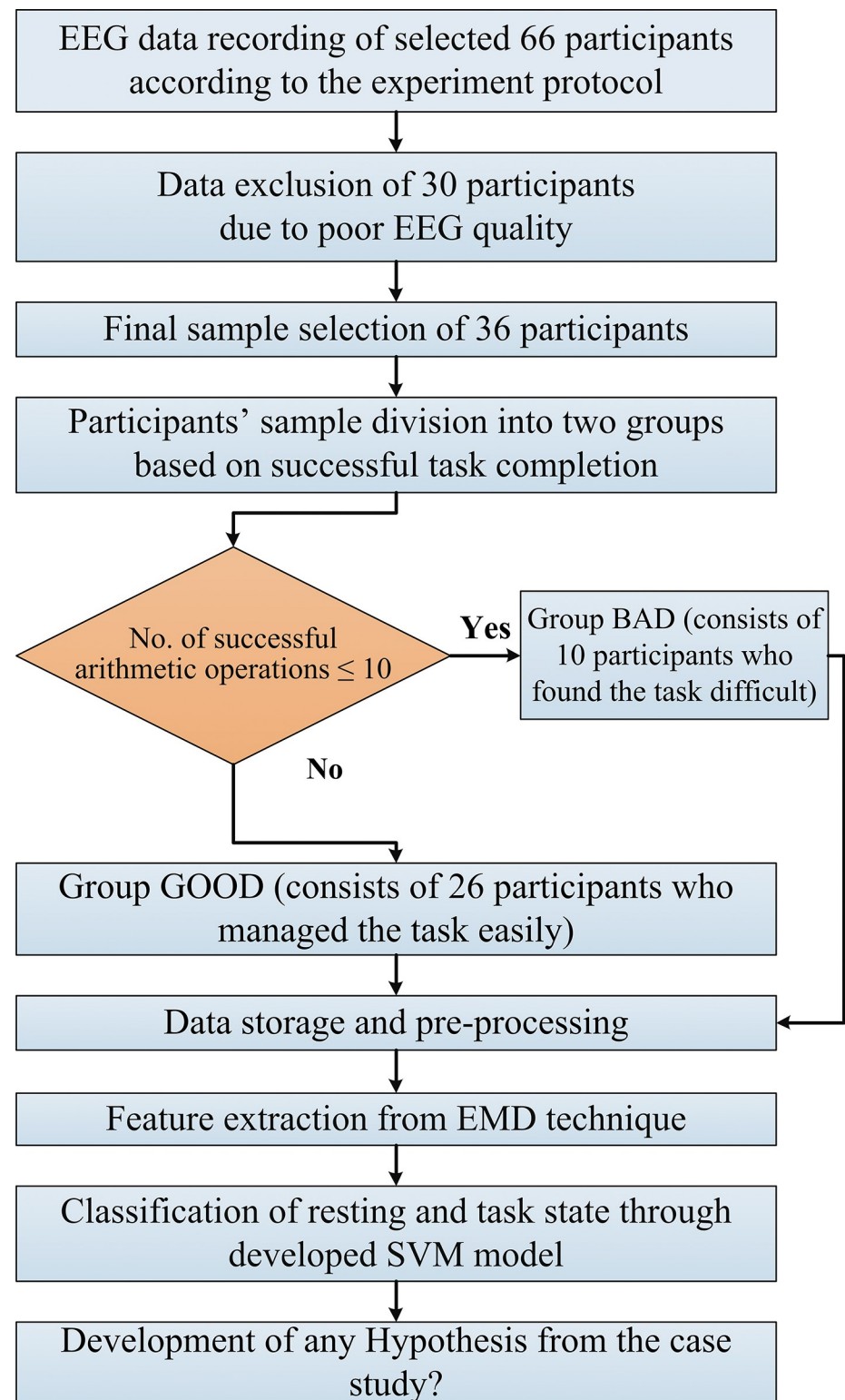

**Fig 2. The complete experimental procedure at a glance.**

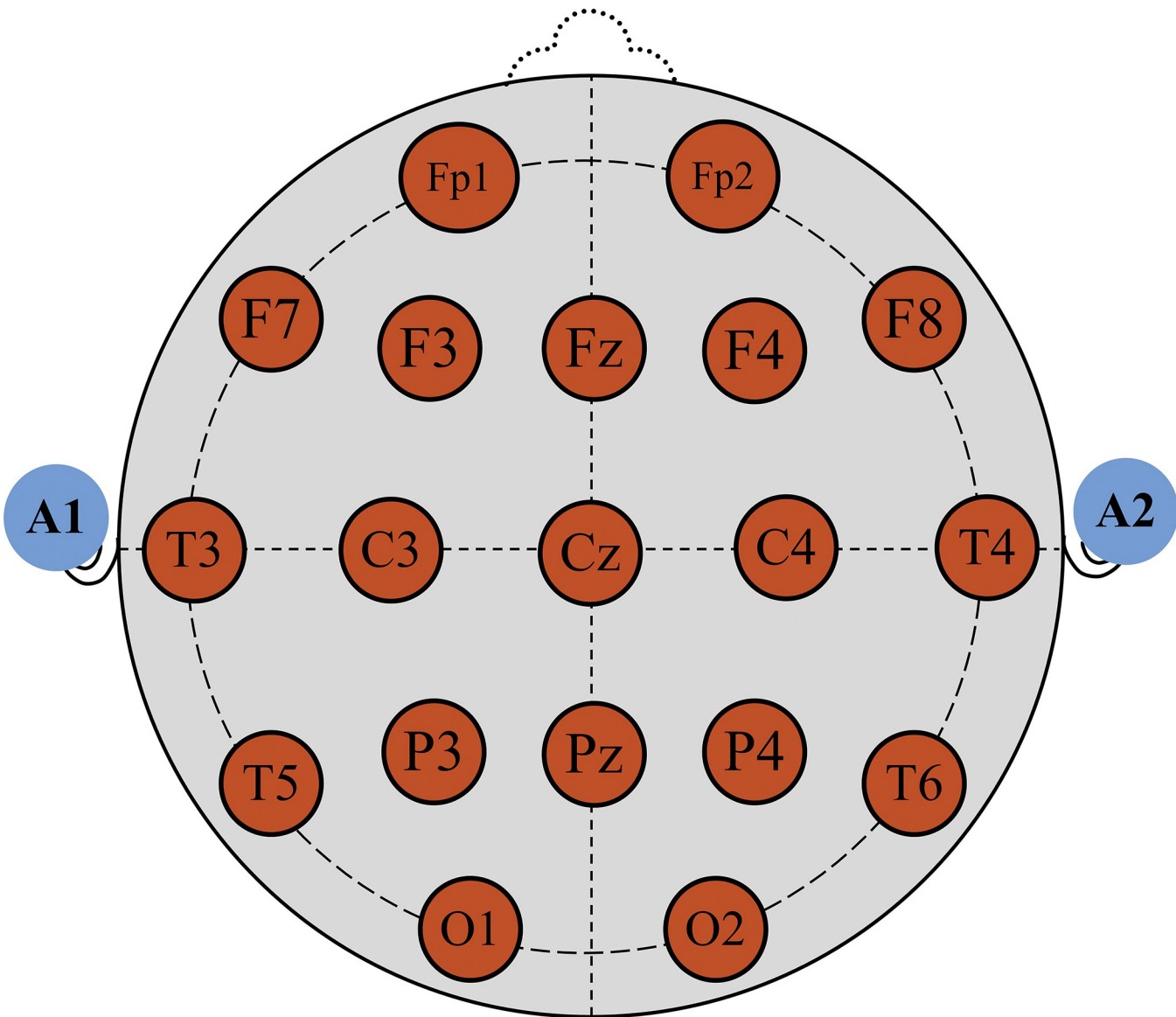

**Fig 3. The EEG electrode placement layout of the data acquisition protocol.**

pattern by decomposing it into a finite sum of intrinsic mode functions (IMF) combined with the Hilbert–Huang transforms (HHT) [39, 40]. EMD is data-adaptive and does not require any hand-crafted functions as well as it extracts the maximum characteristics of EEG over different frequency sub-bands. EMD technique is applied to decompose the EEG signals for preserving informative feature extraction which will enhance the classifier performance. Suppose, EMD iteratively decomposes an $n$-point EEG signal, $x(t)$ to determine IMF mode. Each mode must satisfy the following two conditions defined by the HHT [41].

Condition 1: The number of local extrema and zero crossings must be the same or differ at most by one.

Condition 2: At any point, the mean value of the envelope defined by the local maxima and the envelope by the local minima is zero.

The progression of the basic EMD decomposition algorithm is illustrated in Fig 4.

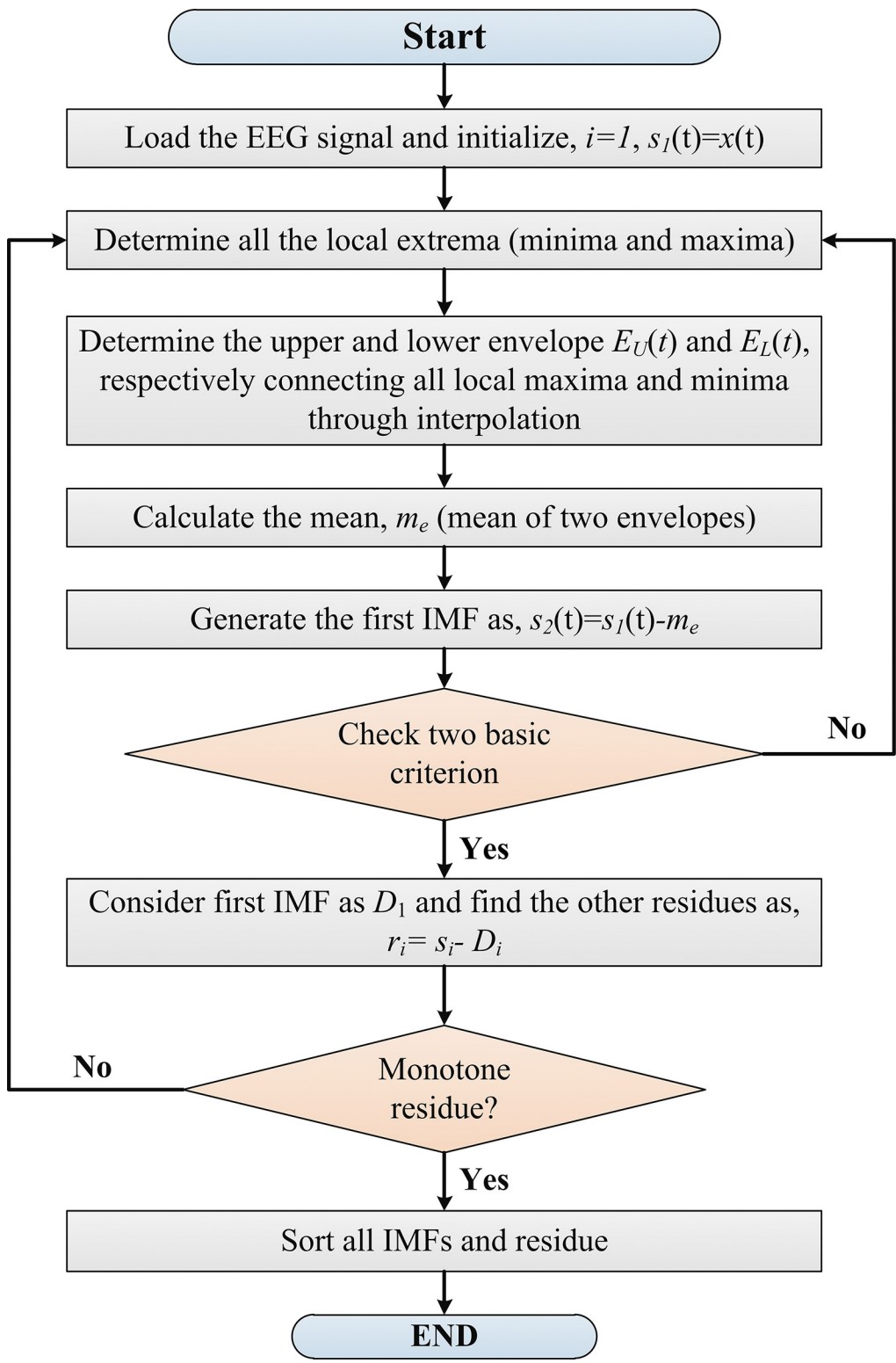

**Fig 4. The flowchart of extracting IMF features through a basic EMD algorithm.**

In the figure, the first decision-making block follows the standard deviation criterion ($SD_{criterion}$) to obtain the IMFs. Apart from this criterion, other stoppage criteria are available such as the $S$ number criterion, threshold method, etc. The $SD_{criterion}$ is equated in (1).

$$SD_{criterion} = \frac{\sum |s_2 - s_1|^2}{\sum |s_1|^2} < q, \text{ where } 0.2 \leq q \leq 0.3 \tag{1}$$

Where, $s_2$ is computed by subtracting the local mean curve from $s_1$ as, $s_2 = s_1 - m_e$. At first, the pre-processed resting and task EEG signals are separated according to BAD (10 rest and 10 task sample data) and GOOD (26 rest and 26 task sample data) groups from a total of 36 participants. Additionally, all participants (36 rest and 36 tasks) sample data has been arranged in chronological order in one unit to determine the collective analysis. Then, the data were decomposed into six (6) IMF among which the first five (5) IMFs contain appropriate sub-bands information encompassing the first 19 channels. These IMFs were taken into account for feature extraction. From these IMFs the following statistical features were extracted: mean, standard deviation (SD), skewness, kurtosis, root mean square (RMS), and the ratio of the absolute mean values of adjacent sub-bands. Since from each IMF six features were extracted, the total feature size from one task was 19×5×6 (channel×IMF×feature). These features are the most well-known parameters to perform the fundamental technical analysis of the data. Considering, an IMF of $n$-point as $D = \{D_1, D_2, D_3, \ldots, D_n\}$, the mean feature calculates the arithmetic average of a set of points or values. So, the mean, $m$ is expressed in (2).

$$m = \frac{1}{n} \sum_{i=1}^{n} D_i \tag{2}$$

The feature, SD ($m_d$) measures the dispersion of data nearby the mean value can be expressed in (3).

$$m_d = \sqrt{\frac{1}{n} \sum_{i=1}^{n} (D_i - m)^2} \tag{3}$$

Additionally, skewness is the third standardized moment which measures the asymmetry of a random variable's distribution around its mean. The skewness ($m_s$) is equated in (4).

$$m_s = \frac{1}{n} \sum_{i=1}^{n} \left(\frac{D_i - m}{m_d}\right)^3 \tag{4}$$

Kurtosis is determined for estimating the relative frequency of extreme values. In one word, it describes the tailedness of a random variable's distribution which follows as in (5),

$$m_k = \frac{1}{n} \sum_{i=1}^{n} \left(\frac{D_i - m}{m_d}\right)^4 \tag{5}$$

Following the RMS feature in mathematics, the root means square measurements of a set) of numbers are calculated as given in (6).

$$RMS = \sqrt{\frac{1}{n} \sum_{i=1}^{n} D_i^2} \tag{6}$$

The RMS value of a signal can be approximated by calculating the RMS of sample data consisting of equally spaced observations. Furthermore, RMS as a statistic feature is used in a

random process; the expected value of RMS is used instead of the mean. Finally, the ratio of the absolute mean values of coefficients of adjacent sub-bands ($R_a$) can be expressed in (7) if we consider $E$ as the values of adjacent sub-bands.

$$R_a = \frac{\sum_{i=1}^{n} |D_i|}{\sum_{i=1}^{n} |E_i|} \tag{7}$$

The extracted features from EMD are used to train the machine learning model and are compared through their classification results. Besides, these features are used to identify the neural correlation with the performance during different cognitive loads. To realize the usefulness of the extracted features based on different types of tasks, the measurement of classification accuracy is obligatory to identify cognitive load.

For classification purposes, several classifiers are initially deployed such as k-Nearest Neighbor, Naive Bayes, Decision Tree, SVM, and Linear Discriminant Analysis. However, SVM is used for the rest analysis of this work, as initially, SVM showed the highest performance. It is also found that SVM is widely accepted in EEG data classification. SVM is an easily embedded supervised machine learning model, popularly used for multi-dimensional feature mapping including accepted prediction accuracy. By definition, SVM builds the maximum distance between the separating hyper-planes and the nearest training points, where the nearest hyper-planes training points are titled as the support vectors. Linear SVM employs linear regression to fit the support vectors. Suppose, a training dataset of $n$ support vector points is in the form of (8).

$$Y = \{(x_1, y_1), (x_2, y_2), \ldots \ldots, (x_n, y_n)\} \tag{8}$$

Here, $y$ will be either 1 or -1 indicating the class of belonging $x$ point. To estimate the maximum-margin hyper-plane, the points of two groups ($x_i$ for $y_i = +1 \, or \, -1$) are set in such a way that the distance between the hyper-planes and the support vectors gets maximized [42]. The hyper-plane can be equated as,

$$w^T x - b = 0 \tag{9}$$

Where $w$ denotes the normal vector on the hyper-plane and the $b$ is related to the offset value as explained in Fig 5. This approach can be extended to other non-linear kernel functions like polynomial, radial basis, etc. To train an SVM classifier, the most significant part is to pick out the appropriate kernel function. In this work, the SVM classifier template was modeled through a 'standardized' predictor and 'polynomial' kernel function with the order of 4 (four). To train the model, fit classification error-correcting output code ('fitcecoc' in MATLAB) is used as binary 'learners' in SVM. A brief graphical explanation of SVM-based classification for two class problems is given in Fig 5.

## Results and discussion

The participants had two mental states: rest and task in this dataset. Therefore, the EEG signals are broadly classified into two classes: rest (class 1) and task (class 2). In the task period, all participants didn't perform their tasks equally. They were to solve mathematical problems (subtraction) within a time limit. It is found that the performance varied from 1 to 30 problems completion on average. The predictive models for mental state classification, features were used to train the model with the first 20 subjects' data, and the rest 16 data were reserved for

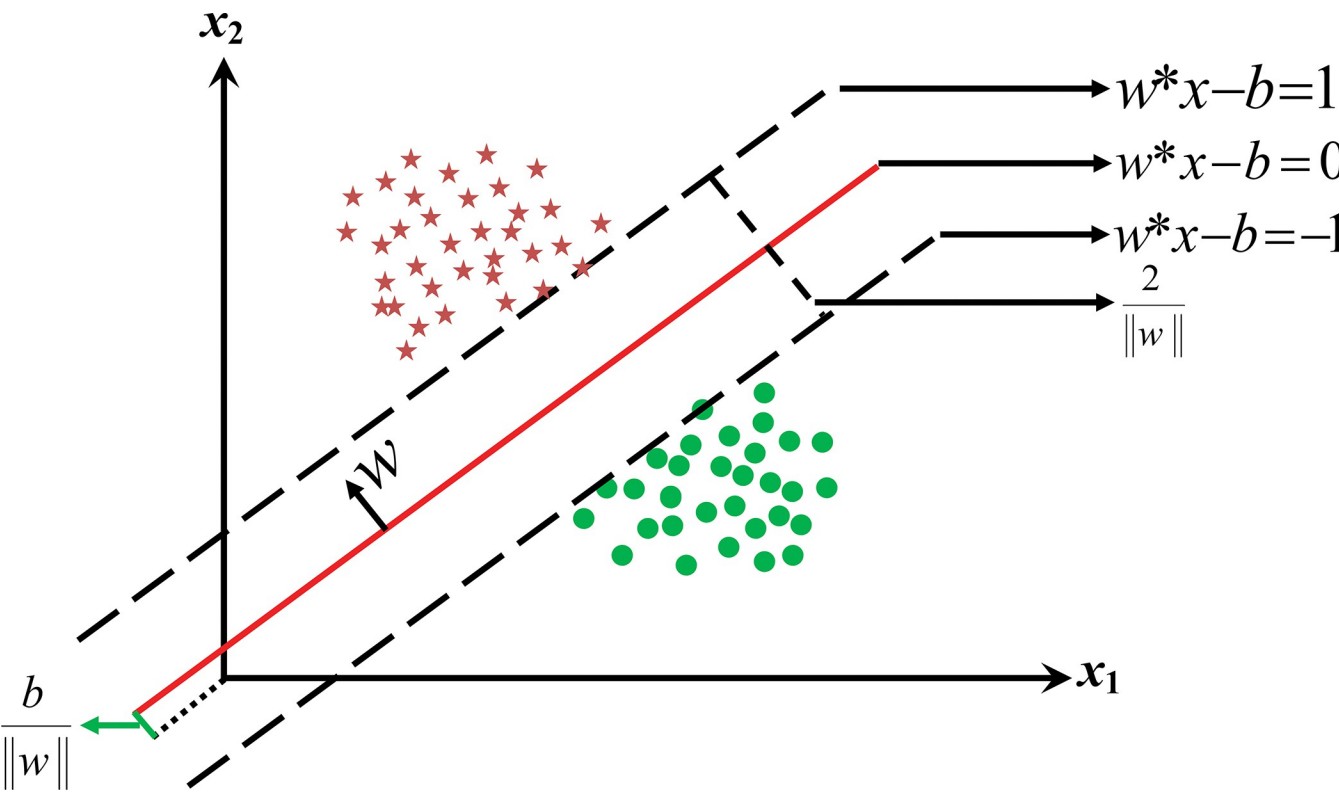

**Fig 5. Hyper-plane, maximum-margin, and support vectors for a linear SVM for two-class classification.**

testing purposes. With the predictive model SVM, EEG signals of these two mental states are classified and found poor results (both classes are classified 37.50% correctly, class 1 is classified 56.25% correctly, class 2 is classified 81.25%, and overall classification accuracy was 68.75%), although it was expected to achieve higher classification accuracy. Some other classifiers were also used to check their performances in this condition such as k-nearest neighbor, decision tree, and linear discriminant analysis. The results are given in Table 1. The classification accuracy of other classifiers is also poor and less than the SVM.

To check the overall performance of the SVM model, all participants' two class data were classified. The detailed classification results of two classes among all participants (36) are given in Table 2. From this result, it was found that accurately both classes were classified as 41.67%, and class 1 (rest) and class 2 (task) were separately classified as 58.33% and 80.8%, respectively. The overall classification accuracy was 71.4%.

To understand this issue more critically, the class prediction states are presented in Tables 3 and 4; where the classification results of GOOD and BAD performers are tabulated separately.

**Table 1. Classification accuracy comparison of some classifiers.**

| S. N. | Classifiers | Train: Test | Classification Accuracy (Both Class) (%) | Overall Classification Accuracy (%) |
|---|---|---|---|---|
| 1 | Linear Discriminant Analysis | 20:16 | 37.50 | 62.50 |
| 2 | SVM | 20:16 | 37.50 | 68.75 |
| 3 | k-Nearest Neighbour | 20:16 | 25.00 | 56.25 |
| 4 | Decision Tree | 20:16 | 31.25 | 62.50 |
| 5 | Naive Bayes | 20:16 | 25.00 | 59.37 |

**Table 2. Classification accuracy between the neural stimulation of rest and task conditions of all participants.**

| Subject ID | Problem Completion (X) | Performance X≤10 = Bad X>10 = Good | Classification Results | | Comments |
|---|---|---|---|---|---|
| | | | Class 1(Rest) | Class 2(Task) | |
| Subject 01 | 9.7 | BAD | **'class_1'** | **'class_2'** | |
| Subject 02 | 29.35 | GOOD | **'class_1'** | **'class_2'** | |
| Subject 03 | 12.88 | GOOD | 'class_1' | 'class_1' | Both are Incorrect as Class 1 |
| Subject 04 | 31 | GOOD | **'class_1'** | **'class_2'** | |
| Subject 05 | 8.6 | BAD | **'class_1'** | **'class_2'** | |
| Subject 06 | 20.71 | GOOD | **'class_1'** | **'class_2'** | |
| Subject 07 | 4.35 | BAD | **'class_1'** | **'class_2'** | |
| Subject 08 | 13.38 | GOOD | 'class_2' | 'class_2' | Both are Incorrect as Class 2 |
| Subject 09 | 18.24 | GOOD | **'class_1'** | **'class_2'** | |
| Subject 10 | 7 | BAD | **'class_1'** | **'class_2'** | |
| Subject 11 | 1 | BAD | 'class_2' | 'class_1' | Reversed |
| Subject 12 | 26 | GOOD | 'class_2' | 'class_2' | Both are Incorrect as Class 2 |
| Subject 13 | 26.36 | GOOD | 'class_1' | 'class_1' | Both are Incorrect as Class 1 |
| Subject 14 | 34 | GOOD | 'class_2' | 'class_2' | Both are Incorrect as Class 2 |
| Subject 15 | 9 | BAD | 'class_2' | 'class_2' | Both are Incorrect as Class 2 |
| Subject 16 | 22.18 | GOOD | 'class_1' | 'class_1' | Both are Incorrect as Class 1 |
| Subject 17 | 11.59 | GOOD | 'class_2' | 'class_2' | Both are Incorrect as Class 2 |
| Subject 18 | 28.7 | GOOD | 'class_2' | 'class_2' | Both are Incorrect as Class 2 |
| Subject 19 | 20 | GOOD | 'class_2' | 'class_2' | Both are Incorrect as Class 2 |
| Subject 20 | 7.06 | BAD | **'class_1'** | **'class_2'** | |
| Subject 21 | 15.41 | GOOD | 'class_2' | 'class_2' | Both are Incorrect as Class 2 |
| Subject 22 | 1 | BAD | **'class_1'** | **'class_2'** | |
| Subject 23 | 4.47 | BAD | **'class_1'** | **'class_2'** | |
| Subject 24 | 27.47 | GOOD | 'class_2' | 'class_2' | Both are Incorrect as Class 2 |
| Subject 25 | 14.76 | GOOD | 'class_2' | 'class_2' | Both are Incorrect as Class 2 |
| Subject 26 | 30.53 | GOOD | 'class_1' | 'class_1' | Both are Incorrect as Class 1 |
| Subject 27 | 13.59 | GOOD | 'class_2' | 'class_2' | Both are Incorrect as Class 2 |
| Subject 28 | 34.59 | GOOD | **'class_1'** | **'class_2'** | |
| Subject 29 | 27 | GOOD | **'class_1'** | **'class_2'** | |
| Subject 30 | 16.59 | GOOD | 'class_1' | 'class_1' | Both are Incorrect as Class 1 |
| Subject 31 | 10 | BAD | 'class_2' | 'class_2' | Both are Incorrect as Class 2 |
| Subject 32 | 19.88 | GOOD | 'class_2' | 'class_2' | Both are Incorrect as Class 2 |
| Subject 33 | 13 | GOOD | **'class_1'** | **'class_2'** | |
| Subject 34 | 21.47 | GOOD | 'class_2' | 'class_2' | Both are Incorrect as Class 2 |
| Subject 35 | 31 | GOOD | 'class_1' | 'class_1' | Both are Incorrect as Class 1 |
| Subject 36 | 12.18 | GOOD | **'class_1'** | **'class_2'** | |
| **Total = 36** | Good Performer = 26 | | Both Class Classification Accuracy = **41.67%** | | Incorrect as Reverse = **01** |
| | Bad Performer = 10 | | Class 1 (Rest) Accuracy = 58.33% | | Both are Incorrect as Class 2 = **14** |
| | | | Class 2 (Task) Accuracy = 80.8% | | Both are Incorrect as Class 1 = **06** |
| | | | Overall Classification Accuracy = 71.4% | | |

[a]The correct classified classes are marked by **bold font**

**Table 3. Classification results and findings of participants based on BAD performance (X≤10).**

| Subject ID | Problem Completion (X) | Classification Results | | Comments |
|---|---|---|---|---|
| | | Class 1(Rest) | Class 2(Task) | |
| Subject 01 | 9.7 | **'class_1'** | **'class_2'** | |
| Subject 05 | 8.6 | **'class_1'** | **'class_2'** | |
| Subject 07 | 4.35 | **'class_1'** | **'class_2'** | |
| Subject 10 | 7 | **'class_1'** | **'class_2'** | |
| Subject 11 | 1 | 'class_2' | 'class_1' | Reversed |
| Subject 15 | 9 | 'class_2' | 'class_2' | Both are Incorrect as Class 2 |
| Subject 20 | 7.06 | **'class_1'** | **'class_2'** | |
| Subject 22 | 1 | **'class_1'** | **'class_2'** | |
| Subject 23 | 4.47 | **'class_1'** | **'class_2'** | |
| Subject 31 | 10 | 'class_2' | 'class_2' | Both are Incorrect as Class 2 |
| Total BAD performer = 10 | X<5 = 04 | Both Class Classification Accuracy = **70%** | | Reversed = **01** |
| | X≥5 = 06 | Class 1 (Rest) Accuracy = 70% | | Both are Incorrect as Class 1 = **00** |
| | | Class 2 (Task) Accuracy = 90% | | Both are Incorrect as Class 2 = **02** |
| | | Overall Classification Accuracy = 80% | | |

[a]The correct classified classes are marked by **bold font**

Here, some interesting findings are observed. The classification accuracy of the BAD performers (both classes classified correctly) is 70% (7 out of 10) where the resting state is classified as 70% and the task state is classified as 90% accurately.

On the other hand, the classification accuracy of the GOOD performers (both classes classified correctly) is only 30.77% (8 out of 26) where the resting state is classified as 53.84% and the task state is classified as 76.92% accurately. Accurately predicted results are marked in green color. If only the overall classification accuracy is observed, some facts remain undercover. It must be noticed that the resting state classification accuracy for the GOOD performers is very low (53.8%) compared to the BAD performers. Besides, this hypothesis can be extended for the task state classification too because here the task signals of BAD performers have been classified 90% accurately while this rate of the GOOD performers is 76.92%. Since we generally hypothesize that the EEG signal pattern of the task state creates significant variation compared to the EEG signal of the resting state, such poor results are not expected. What is the problem?

This work hypothesized some facts that can answer the raised question regarding the issue:

➢ Firstly, it is assumed that the participants who performed very well in mathematical problem-solving might face no mental stress significantly.

➢ Secondly, the BAD performers faced strong enough stress during task state except those who performed badly at marginal level (solved 9~10 math problems).

➢ Thirdly, this complementary response of the rest and task pattern of the EEG signal influenced the learning state (training step) of the SVM which produced a wrong predictive model.

These three factors eventually reduced the classification accuracy of the EEG signal related to the mental rest and task states. If these three factors can be fixed, the classification accuracy may be increased. Therefore, after fixing the issues (based on the proposed hypotheses) if the classification accuracy is increased, the hypotheses will be proved.

**Table 4. Classification results and findings of participants based on GOOD performance (X>10).**

| Subject ID | Problem Completion (X) | Classification Results | | Comments |
|---|---|---|---|---|
| | | Class 1(Rest) | Class 2(Task) | |
| Subject 02 | 29.35 | **'class_1'** | **'class_2'** | |
| Subject 03 | 12.88 | 'class_1' | 'class_1' | Both are Incorrect as Class 1 |
| Subject 04 | 31 | **'class_1'** | **'class_2'** | |
| Subject 06 | 20.71 | **'class_1'** | **'class_2'** | |
| Subject 08 | 13.38 | 'class_2' | 'class_2' | Both are Incorrect as Class 2 |
| Subject 09 | 18.24 | **'class_1'** | **'class_2'** | |
| Subject 12 | 26 | 'class_2' | 'class_2' | Both are Incorrect as Class 2 |
| Subject 13 | 26.36 | 'class_1' | 'class_1' | Both are Incorrect as Class 1 |
| Subject 14 | 34 | 'class_2' | 'class_2' | Both are Incorrect as Class 2 |
| Subject 16 | 22.18 | 'class_1' | 'class_1' | Both are Incorrect as Class 1 |
| Subject 17 | 11.59 | 'class_2' | 'class_2' | Both are Incorrect as Class 2 |
| Subject 18 | 28.7 | 'class_2' | 'class_2' | Both are Incorrect as Class 2 |
| Subject 19 | 20 | 'class_2' | 'class_2' | Both are Incorrect as Class 2 |
| Subject 21 | 15.41 | 'class_2' | 'class_2' | Both are Incorrect as Class 2 |
| Subject 24 | 27.47 | 'class_2' | 'class_2' | Both are Incorrect as Class 2 |
| Subject 25 | 14.76 | 'class_2' | 'class_2' | Both are Incorrect as Class 2 |
| Subject 26 | 30.53 | 'class_1' | 'class_1' | Both are Incorrect as Class 1 |
| Subject 27 | 13.59 | 'class_2' | 'class_2' | Both are Incorrect as Class 2 |
| Subject 28 | 34.59 | **'class_1'** | **'class_2'** | |
| Subject 29 | 27 | **'class_1'** | **'class_2'** | |
| Subject 30 | 16.59 | 'class_1' | 'class_1' | Both are Incorrect as Class 1 |
| Subject 32 | 19.88 | 'class_2' | 'class_2' | Both are Incorrect as Class 2 |
| Subject 33 | 13 | **'class_1'** | **'class_2'** | |
| Subject 34 | 21.47 | 'class_2' | 'class_2' | Both are Incorrect as Class 2 |
| Subject 35 | 31 | 'class_1' | 'class_1' | Both are Incorrect as Class 1 |
| Subject 36 | 12.18 | **'class_1'** | **'class_2'** | |
| Total GOOD performer = 26 | X<20 = 11 | Both Class Classification Accuracy = **30.77%** | | Reversed = **00** |
| | X≥20 = 15 | Class 1 (Rest) Accuracy = 53.84% | | Both are Incorrect as Class 1 = **06** |
| | | Class 2 (Task) Accuracy = 76.92% | | Both are Incorrect as Class 2 = **12** |
| | | Overall Classification Accuracy = 66.88% | | |

[a]The correct classified classes are marked by **bold font**

To establish the hypotheses, we have investigated some important facts. Estimated the results from Table 5, the significant information we can perceive that training the rest and task state from the BAD performers' data created such a machine learning model that can classify accurately (100%) the resting states of mind of GOOD performers' data. Contrariwise, the model fully failed to detect task state (classified all as class_1 instead of class_2) for all GOOD performers. It means the significant difference between task and rest was modeled for the training data of BAD performers but doesn't work for the task data of GOOD performers. Based on this result, it can be assumed that those who performed better didn't feel strong enough mental stress due to their skill. This result supports our first hypothesis.

Following this outcome, we did another experiment considering only the GOOD performers. We wanted to see what happens between the rest and task states among the GOOD performance. In that case, GOOD performers were organized in ascending order regarding their performances. Among 26 participants in ascending order of GOOD performances, the first 10

**Table 5. Classification performance while conditions were set as trained with the BAD performers' data (10 participants) and test with the GOOD performers' data (26 participants).**

| Subject ID | Problem Completion | Classification Results | | Subject ID | Problem Completion | Classification Results | |
|---|---|---|---|---|---|---|---|
| | | Class 1 (Rest) | Class 2 (Task) | | | Class 1 (Rest) | Class 2 (Task) |
| Subject 02 | 29.35 | 'class_1' | **'class_1'** | Subject 21 | 15.41 | 'class_1' | **'class_1'** |
| Subject 03 | 12.88 | 'class_1' | **'class_1'** | Subject 24 | 27.47 | 'class_1' | **'class_1'** |
| Subject 04 | 31 | 'class_1' | **'class_1'** | Subject 25 | 14.76 | 'class_1' | **'class_1'** |
| Subject 06 | 20.71 | 'class_1' | **'class_1'** | Subject 26 | 30.53 | 'class_1' | **'class_1'** |
| Subject 08 | 13.38 | 'class_1' | **'class_1'** | Subject 27 | 13.59 | 'class_1' | **'class_1'** |
| Subject 09 | 18.24 | 'class_1' | **'class_1'** | Subject 28 | 34.59 | 'class_1' | **'class_1'** |
| Subject 12 | 26 | 'class_1' | **'class_1'** | Subject 29 | 27 | 'class_1' | **'class_1'** |
| Subject 13 | 26.36 | 'class_1' | **'class_1'** | Subject 30 | 16.59 | 'class_1' | **'class_1'** |
| Subject 14 | 34 | 'class_1' | **'class_1'** | Subject 32 | 19.88 | 'class_1' | **'class_1'** |
| Subject 16 | 22.18 | 'class_1' | **'class_1'** | Subject 33 | 13 | 'class_1' | **'class_1'** |
| Subject 17 | 11.59 | 'class_1' | **'class_1'** | Subject 34 | 21.47 | 'class_1' | **'class_1'** |
| Subject 18 | 28.7 | 'class_1' | **'class_1'** | Subject 35 | 31 | 'class_1' | **'class_1'** |
| Subject 19 | 20 | 'class_1' | **'class_1'** | Subject 36 | 12.18 | 'class_1' | **'class_1'** |

Individual Accuracy: Class_1 = **100%** and Class_2 = **0.00%**
Total Accuracy = **50%**

[a]The wrong classified classes are marked by **bold font**

participants were considered Average_GOOD performers, and the rest 16 participants were considered GOOD performers. Therefore upon this categorization, GOOD performers were split into two categories: Average_GOOD and GOOD. Now, with the data of Average_GOOD performers were taken for training a machine learning model, and eventually, with the help of this model, data of GOOD performers (the rest 16 participants) were tested and the classification result is given in Table 6. From this consideration, the output we get is more interesting; because the learned model got confused to differentiate between two classes. Among the 16 participants' data of GOOD performers, the SVM model confused to differentiate two classes 15 times. These results are marked in blue color. In detail, the resting state data of Average_GOOD participants have failed to identify GOOD participants' rest state data which reflects

**Table 6. Classification performance while conditions were set as trained with the Average_GOOD performers' data (10 participants) and test with the GOOD performers' data (16 participants).**

| Subject ID | Problem Completion | Classification Results | | Subject ID | Problem Completion | Classification Results | |
|---|---|---|---|---|---|---|---|
| | | Class 1 (Rest) | Class 2 (Task) | | | Class 1 (Rest) | Class 2 (Task) |
| Subject 32 | 29.35 | **'class_2'** | 'class_2' | Subject 24 | 27.47 | 'class_1' | **'class_1'** |
| Subject 19 | 12.88 | 'class_1' | 'class_2' | Subject 18 | 28.7 | 'class_1' | 'class_2' |
| Subject 06 | 31 | **'class_2'** | 'class_2' | Subject 02 | 29.35 | 'class_1' | 'class_2' |
| Subject 34 | 20.71 | **'class_2'** | **'class_1'** | Subject 26 | 30.53 | **'class_2'** | **'class_1'** |
| Subject 16 | 13.38 | **'class_2'** | **'class_1'** | Subject 04 | 31 | **'class_2'** | 'class_2' |
| Subject 12 | 18.24 | **'class_2'** | 'class_2' | Subject 35 | 31 | 'class_1' | 'class_2' |
| Subject 13 | 26 | **'class_2'** | 'class_2' | Subject 14 | 34 | **'class_2'** | 'class_2' |
| Subject 29 | 26.36 | **'class_2'** | 'class_2' | Subject 28 | 34.59 | **'class_2'** | 'class_2' |

Individual Accuracy: Class_1 = 31.25% and Class_2 = 75%
Total Accuracy = 53.12%

[a]The wrong classified classes are marked by **bold font**

**Table 7. Classification performance while conditions were set as trained with first 18 subjects (including GOOD and BAD) and tested with all 36 subjects.**

| Subject ID | Problem Completion | Classification Results | | Subject ID | Problem Completion | Classification Results | |
|---|---|---|---|---|---|---|---|
| | | Class 1 (Rest) | Class 2 (Task) | | | Class 1 (Rest) | Class 2 (Task) |
| Subject 01 | 9.7 | 'class_1' | 'class_2' | Subject 19 | 20 | 'class_1' | 'class_2' |
| Subject 02 | 29.35 | 'class_1' | 'class_2' | Subject 20 | 7.06 | 'class_1' | 'class_2' |
| Subject 03 | 12.88 | 'class_1' | 'class_2' | Subject 21 | 15.41 | 'class_1' | 'class_2' |
| Subject 04 | 31 | 'class_1' | 'class_2' | Subject 22 | 1 | 'class_1' | 'class_2' |
| Subject 05 | 8.6 | 'class_1' | 'class_2' | Subject 23 | 4.47 | 'class_1' | 'class_2' |
| Subject 06 | 20.71 | 'class_1' | 'class_2' | Subject 24 | 27.47 | 'class_1' | 'class_2' |
| Subject 07 | 4.35 | 'class_1' | 'class_2' | Subject 25 | 14.76 | **'class_2'** | 'class_2' |
| Subject 08 | 13.38 | 'class_1' | 'class_2' | Subject 26 | 30.53 | 'class_1' | **'class_1'** |
| Subject 09 | 18.24 | 'class_1' | 'class_2' | Subject 27 | 13.59 | 'class_1' | 'class_2' |
| Subject 10 | 7 | 'class_1' | 'class_2' | Subject 28 | 34.59 | 'class_1' | 'class_2' |
| Subject 11 | 1 | 'class_1' | 'class_2' | Subject 29 | 27 | 'class_1' | 'class_2' |
| Subject 12 | 26 | 'class_1' | 'class_2' | Subject 30 | 16.59 | 'class_1' | **'class_1'** |
| Subject 13 | 26.36 | 'class_1' | 'class_2' | Subject 31 | 10 | 'class_1' | 'class_2' |
| Subject 14 | 34 | 'class_1' | 'class_2' | Subject 32 | 19.88 | 'class_1' | **'class_1'** |
| Subject 15 | 9 | 'class_1' | 'class_2' | Subject 33 | 13 | 'class_1' | 'class_2' |
| Subject 16 | 22.18 | 'class_1' | 'class_2' | Subject 34 | 21.47 | **'class_2'** | 'class_2' |
| Subject 17 | 11.59 | 'class_1' | 'class_2' | Subject 35 | 31 | 'class_1' | 'class_2' |
| Subject 18 | 28.7 | 'class_1' | 'class_2' | Subject 36 | 12.18 | 'class_1' | 'class_2' |
| Reversed = **00** | | | | Classification Accuracy = 93.0556% | | | |
| Both are Incorrect as 1 = **03** | | | | Class 1 (Rest) Accuracy = 94.4444% | | | |
| Both are Incorrect as 2 = **02** | | | | Class 2 (Task) Accuracy = 91.6667% | | | |

[a]The wrong classified classes are marked by **bold font**

divergence among resting states of GOOD performers. Perhaps, the Average_GOOD performers felt mental stress in resting time also as well as tasking time. This result also partially supports our hypothesis. If we consider the Average_GOOD and BAD performers (suggested by the data acquiring team), our further consideration result should be accurate at the accepted level but the resulting accuracy is found 53.12%.

To achieve acceptable classification accuracy through a logical training process and to justify the hypothesis, the classifying model has been trained with 18 participants (50% of the total population including Average_GOOD and BAD) with rest and task data. This model is tested with all participants (36) to categorize the two classes (rest and task). The classification results are given in Table 7.

It is found that this hypothesis achieves 93.06% accuracy which is pretty good and the individual accuracy is also in an acceptable range. This result means that if we train the proposed model with EMD features of GOOD and BAD performers' data in a particular way, it can classify the data with greater accuracy and strongly supports our hypotheses. Besides, in the same way, the classification accuracy was calculated for different polynomial orders of the kernel function. From the results, it is found that for polynomial order of 4; the classification accuracy was highest. The comparative results are given in Fig 6.

As we categorized the participants into three different groups (Average_GOOD, GOOD, and BAD) regarding their performance, it was expected to visualize the different patterns in EEG signal power. On this general hypothesis, the signal power of all channels is calculated for all participants individually. To calculate the power, the rest and task data are normalized and after calculating the channel-wise signal power, the signal power is grand averaged according

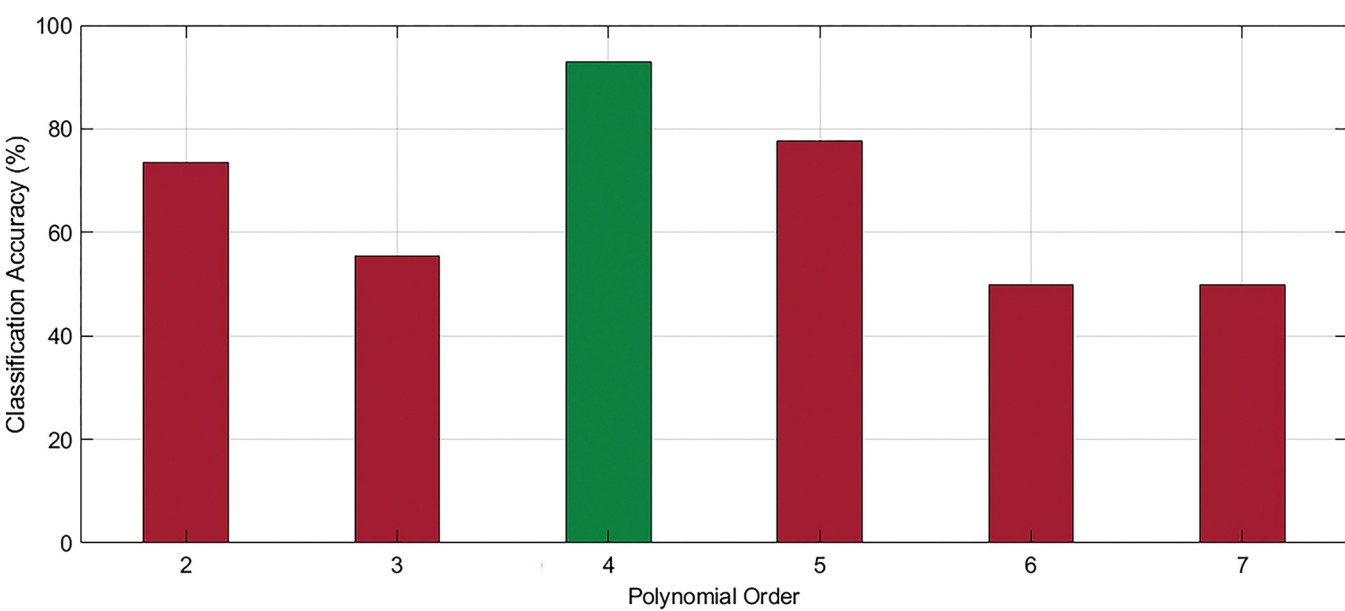

**Fig 6. The comparison of classification accuracy for different polynomial orders of SVM.**

to the category of BAD, Average_GOOD, and GOOD performers, respectively. The channel-wise grand averaged power during rest and the task is given in Figs 7–9, respectively on the proposed categories. In Fig 7, the difference in signal power regarding rest and task is found

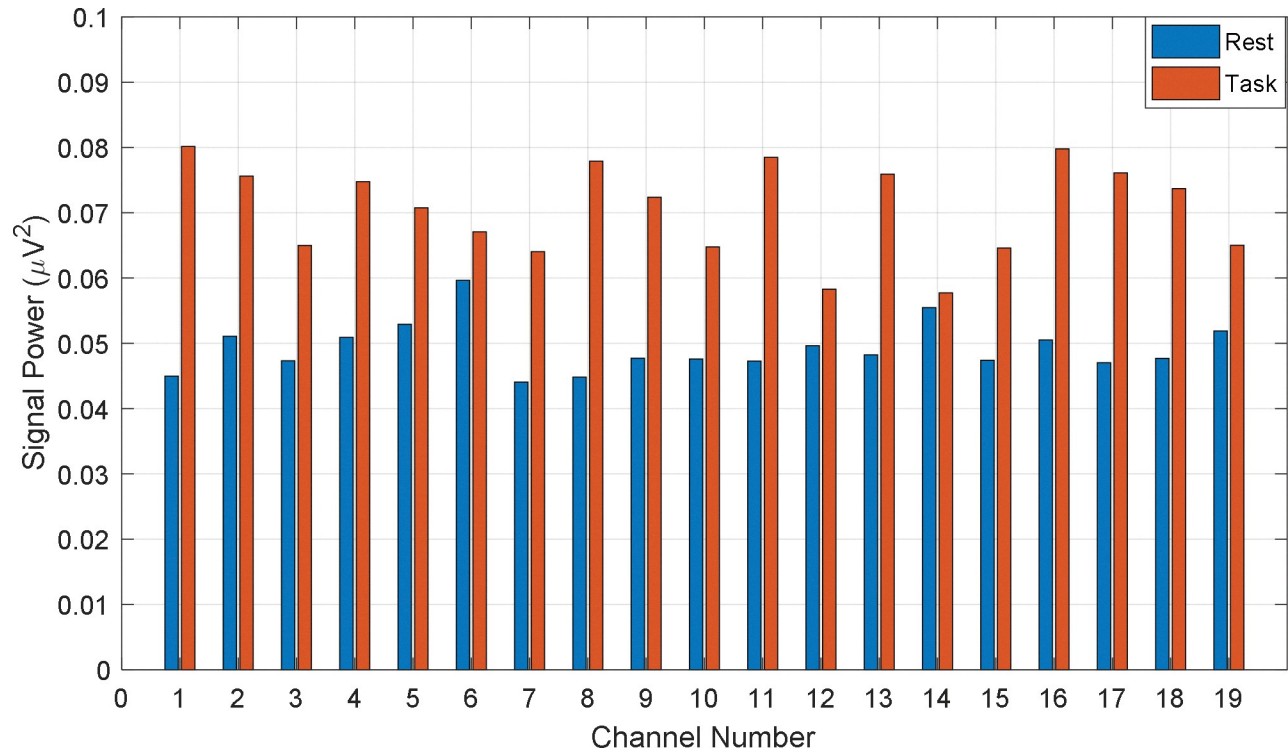

**Fig 7. Channel-wise mean signal power (rest vs. task) from BAD performances.**

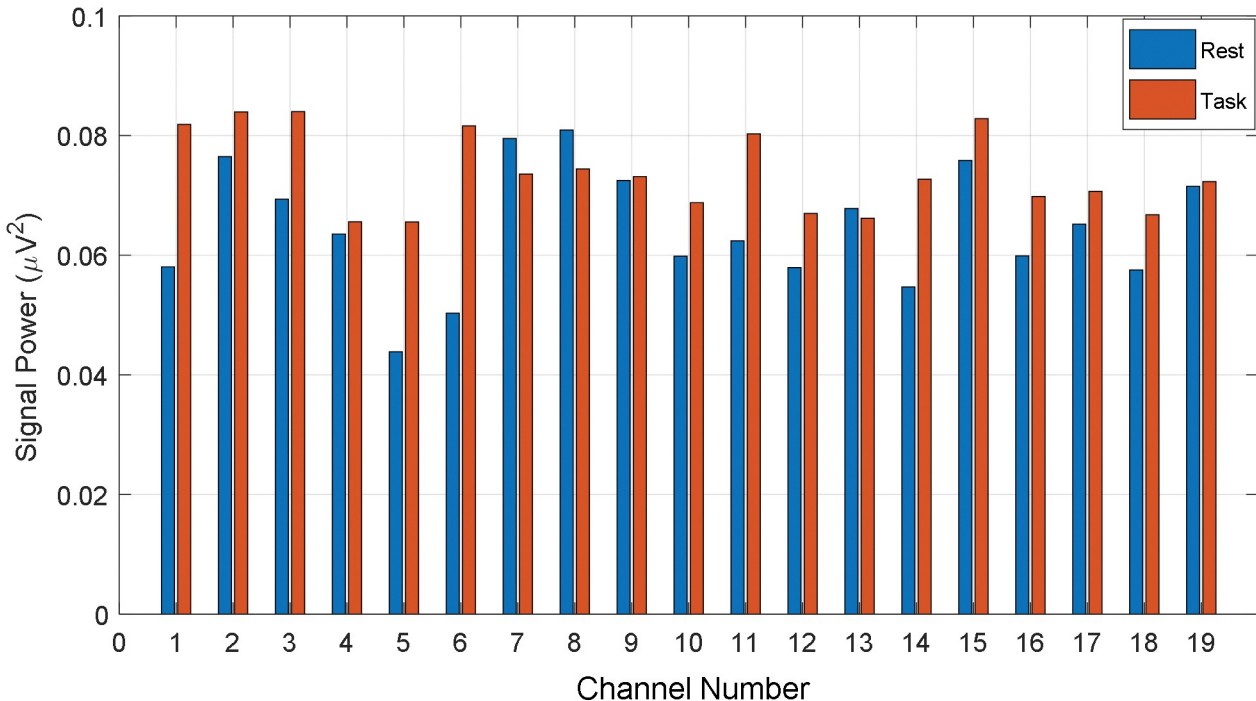

**Fig 8. Channel-wise mean signal power (rest vs. task) from Average_GOOD performances.**

significant. That means BAD performers felt more cognitive load or mental stress during task time rather than during the rest period. Similarly, the channel-wise mean signal powers (rest vs. task) from Average_GOOD and GOOD performances are chronologically presented in Figs 8 and 9. Their fluctuating results are found for Average_GOOD performers which are shown in Fig 8. This result depicts that the variations of signal power due to the rest and task period are random and inconsistent for all channels. In some channels, the mean signal power of the task is greater than the rest condition and in some channels, this result is altered. Besides, in some channels, the powers are almost the same. Therefore, these participants had an exceptional pattern in their signal power during their rest and task period.

The differences in signal power between rest and task period for GOOD performers are visible in Fig 9 (signal power of task period is higher than that of rest) but very insignificant compared to the BAD performers. Therefore, GOOD performers in math problem solutions feel low stress about the task rather than those who are weak in math. In other words, being math experts, the participants did not feel any significant cognitive load during the mental math task. This result supports our hypotheses on the performance-based cognitive load.

Finally, the signal power results are plotted as head topography for BAD, Average_GOOD, and GOOD performers in Figs 10–12, respectively. Similar to the previous bar graph results, the topoplot of BAD performers in Fig 10 has presented that at task period the BAD performers felt high cognitive load at the middle part of the frontal, central, and parietal region of the human brain; whereas at rest period they felt almost no load slightly at the frontal region which is negligible. Alike of bar graph, the Average_GOOD performers have shown confusing results in Fig 11. The figure reveals that Average_GOOD performers felt less cognitive load in performing tasks rather than resting sessions.

Lastly, the GOOD performers in Fig 12 felt the same cognitive load at rest and task period. Because those who are good in math, do not feel any extra cognitive load for the task period as

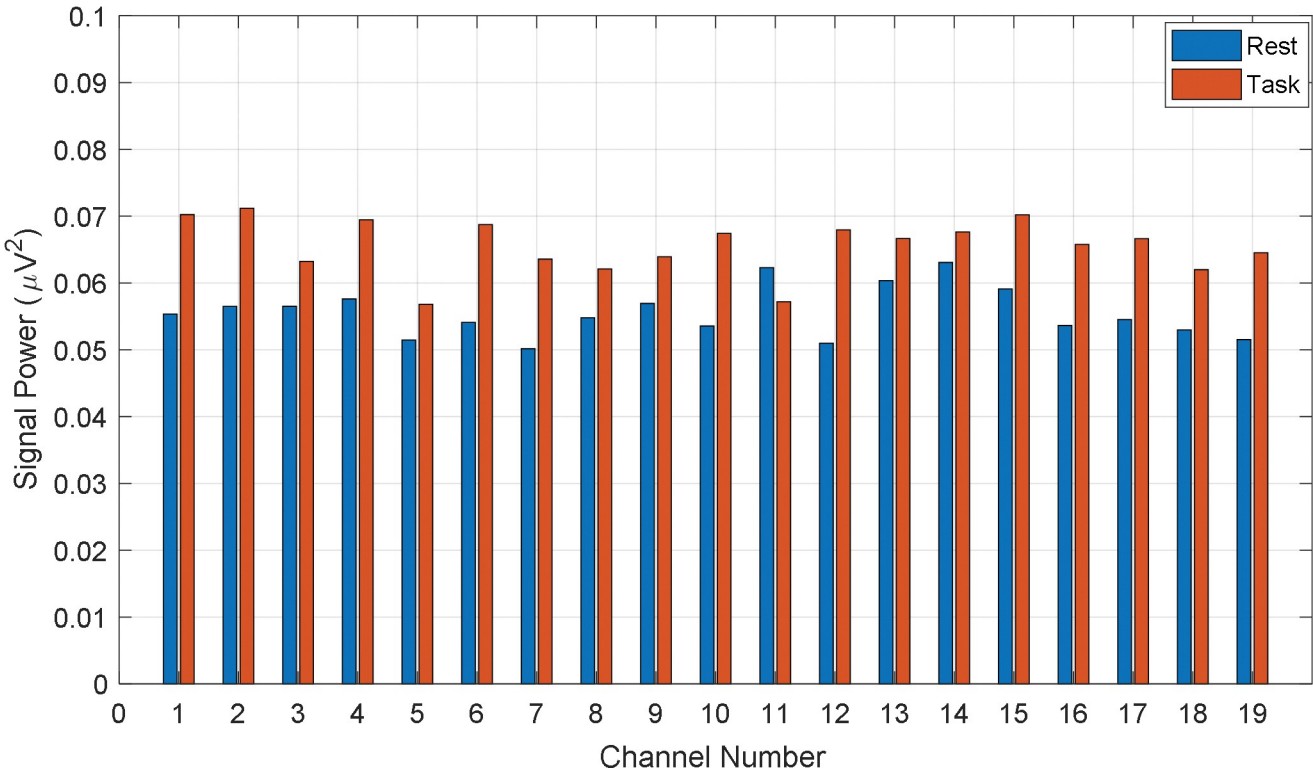

**Fig 9. Channel-wise mean signal power (rest vs. task) from GOOD performances.**

the BAD performers feel. Therefore, unlike the conventional concept of increased mental load during the task than the resting condition, this work revealed that the mental workload can be subjected to an independent manner and depends on task performance.

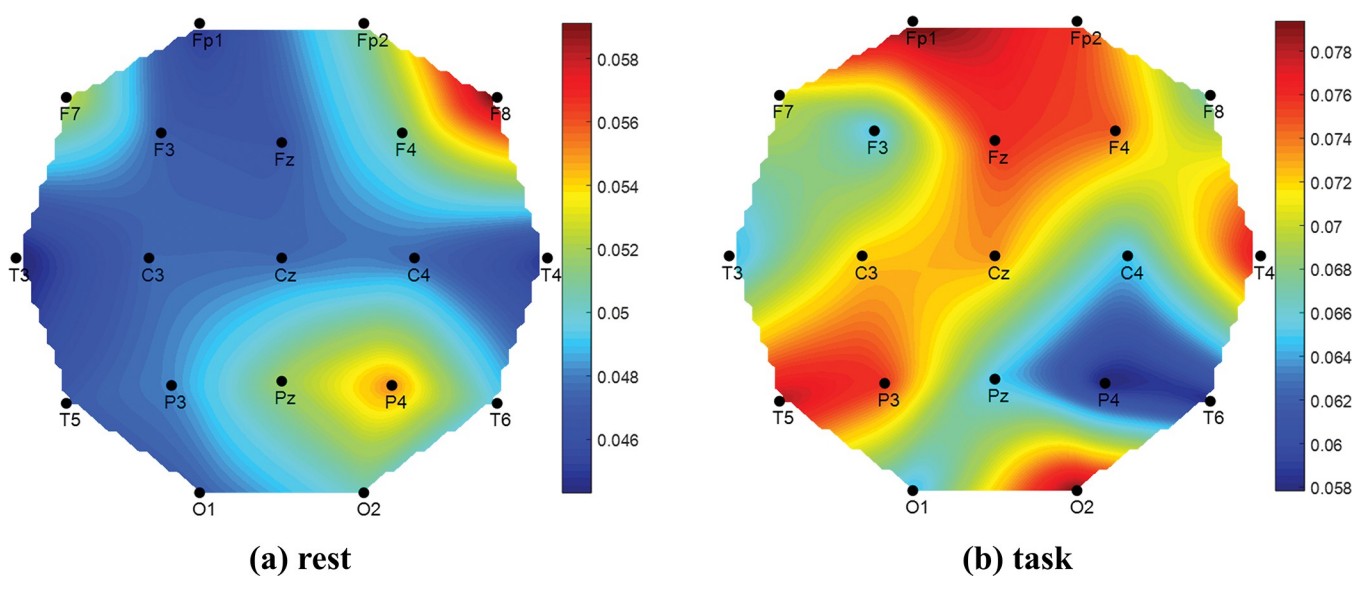

**(a) rest** **(b) task**

**Fig 10. Topography of EEG signal power ($\mu V^2$) for BAD performers.**

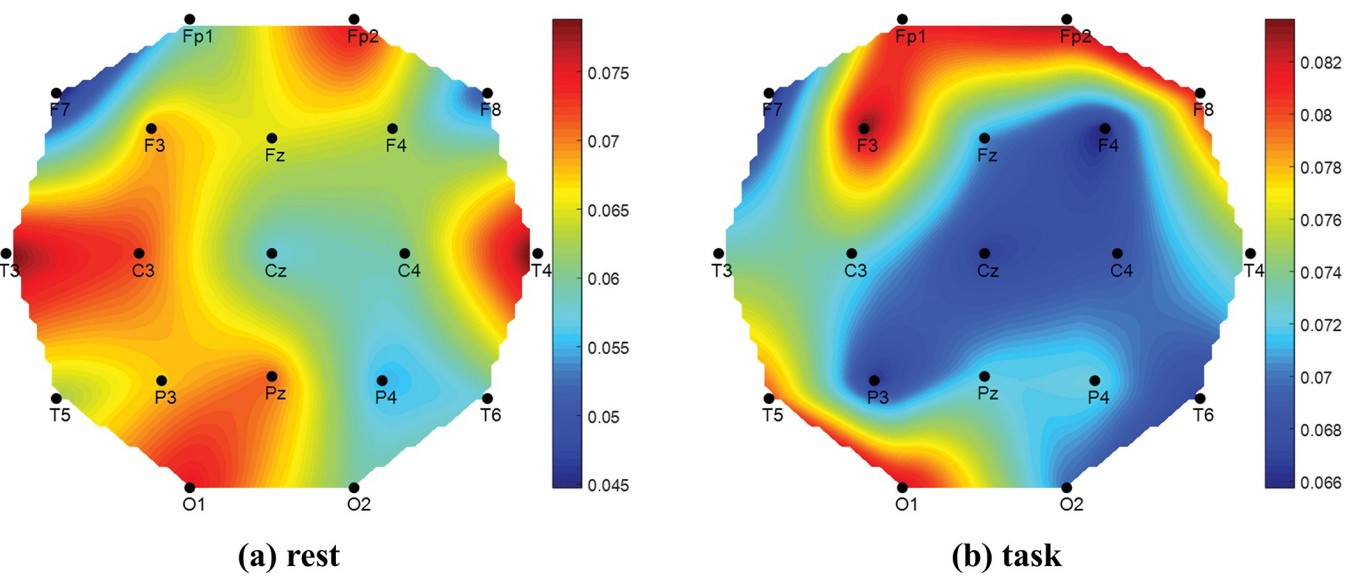

**Fig 11. Topography of EEG signal power (µV²) for Average_GOOD performers.**

Since the higher value of the image similarity index (ISI) indicates the similarity between two images, the similarity of two topoplots can be compared with the value of ISI. The performance-wise groups (BAD, Average_GOOD, and GOOD) are considered to prepare their topoplots as described earlier and compared their ISIs. The results are given in Table 8. From the table, GOOD performers' rest and task topoplot images are more similar than others as their ISI shows higher value. This numerical value again justifies our hypothesis as GOOD performers feel similar mental stress for rest and task conditions.

There is a difference in mental workload or cognitive load between two participants who are performing the given task better than the other. Hence this work indirectly claims that there is a strong correlation between the neural activity during a task and the achieved

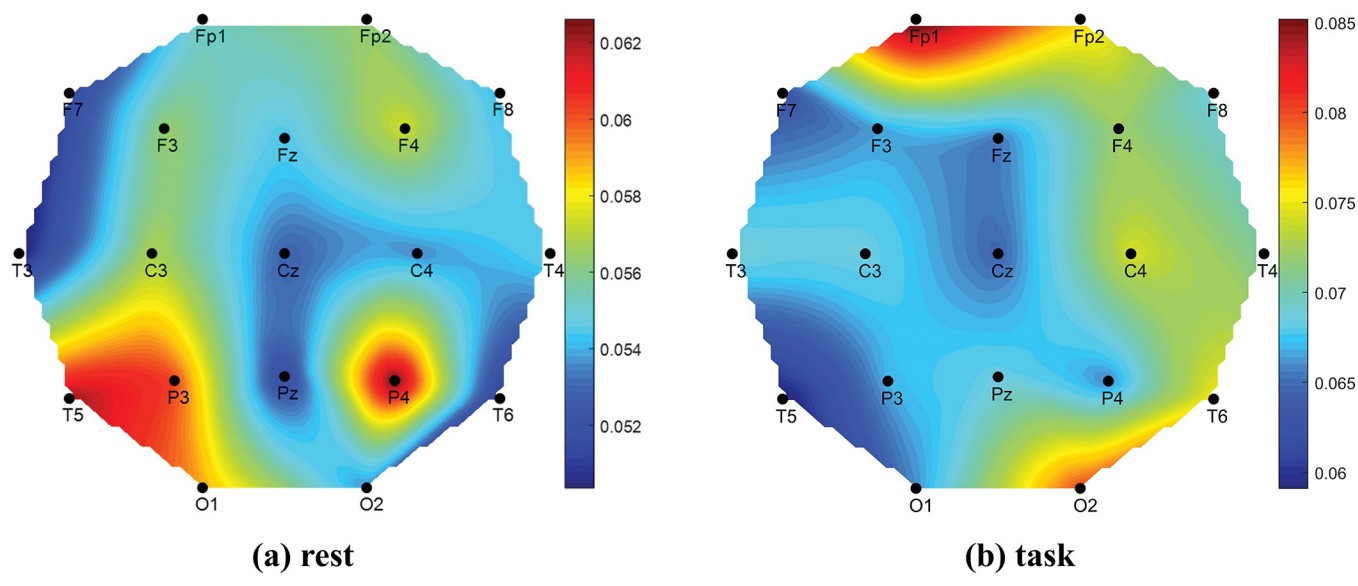

**Fig 12. Topography of EEG signal power (µV²) for GOOD performers.**

**Table 8. Image-to-image similarity check among the topoplots.**

| Comparing Events (Rest vs Task) | BAD | Average_GOOD | GOOD |
|---|---|---|---|
| ISI | 0.4766 | 0.4195 | **0.6499** |

performance of the participant. In other words, the cognitive workload is relevant to the subjects' skills or experience. Expert persons will perform well and get less stress in the brain during particular tasks under their expertise. This result can help the researchers or clinical personnel concerning the cognitive neuroscience field to evaluate the actual workload considering their previous expertise. In addition, this result can also facilitate to design the appropriate stimuli for the participants in the cognition based neuroscientific research.

Alternatively, inexperienced persons will perform inferior to that task and experience considerable stress during the task which results in a higher cognitive load. In the research proposal, it is suggested that a balanced ratio of GOOD and BAD performers' results should be considered for training the model to get satisfactory classification accuracy. The findings of this study and the drawn hypotheses will be useful in relevant research on cognitive load estimation and analysis. However, the hypotheses developed from this research work could be more validated if there are more datasets where the performance of the participants would also be tabulated. Concluding hypotheses on only one dataset may be considered a limitation of this work. Therefore, this result guides future researchers to acquire mental workload-related functional brain signals from EEG, fNIRS, fMRI, MEG, etc. along with their performances (as metadata).

## Conclusions

This research work investigated the neural correlation with cognitive task performance considering the subject-independent approach. The EEG data of thirty-six subjects were acquired during their rest and task conditions. The participants solved the mathematical problem as cognitive tasks and showed different performances regarding their tasks. Some hypotheses are drawn upon the impact of task performance on cognitive load.

To prove the hypothesis, this work has conducted a chain of methodological steps. To determine the cognitive load levels, a renowned EEG dataset of arithmetic task performance including resting period data is chosen. After pre-processing, the whole dataset is grouped into two categories (GOOD and BAD) based on the amount of task performance within a fixed time limit. To classify the cognitive load levels (rest and task); the SVM classifier is trained with the features extracted from the EMD technique. The classifier is modeled with several amounts of train sets of data to discover the supportive evidence in favor of the drawn hypotheses. To get the appropriate judgment, the GOOD performance of data is again divided into Average_GOOD and GOOD performance. It is found that the differences between rest and task levels of BAD and GOOD performers are low and high, respectively because BAD performers felt more cognitive load which supports the hypothesis. The differences between Average_GOOD and GOOD performers' rest and task levels are confusing because all GOOD performers felt the almost same amount of cognitive load at rest and task period. These findings also support the proposed hypothesis which claims the strong correlation between neural activity and task performance for different cognitive load levels based on the expertise or experience of the performers. In accumulation, the topographical presentation of the EEG signal power regarding three types of participants' performance levels is easily comparable in the context of the functional pattern and image-to-image correlation.

The outcome of this research will create a distinct impact in the field of cognitive psychology. To establish the wide acceptance of the proposed research hypothesis, some other datasets

of different cognitive workloads can be considered to study and compare through cognitive load level classification from the EEG signals. Because this type of hypothesis is not attempted yet. To improvise the research idea, a comparative study of participants' experimental time gestures and detailed brain functional patterns should be considered in the future work strategy.

## Supporting information

**S1 Table. Classification accuracy of different polynomial orders in SVM (Numerical data of Fig 6).**
(PDF)

**S2 Table. Normalized channel power for the bad performers (Numerical data of Fig 7).**
(PDF)

**S3 Table. Normalized channel power for the average performers (Numerical data of Fig 8).**
(PDF)

**S4 Table. Normalized channel power for the GOOD performers (Numerical data of Fig 9).**
(PDF)

## Acknowledgments

The authors would like to acknowledge the role of Dr. Md. Asadur Rahman, Assistant Professor, Department of Biomedical Engineering, MIST, Dhaka, Bangladesh for his guidance to accomplish this research work.

The experimental protocol of the used data was approved by the Bioethics Commission of the Educational and Scientific Centre at the "Institute of Biology and Medicine", in the Taras Shevchenko National University of Kyiv conferring to the Helsinki declaration.

## Author Contributions

**Conceptualization:** Farzana Khanam.

**Data curation:** Farzana Khanam.

**Formal analysis:** Farzana Khanam.

**Methodology:** Farzana Khanam.

**Supervision:** Mohiuddin Ahmad, A. B. M. Aowlad Hossain.

**Writing – original draft:** Farzana Khanam.

**Writing – review & editing:** Mohiuddin Ahmad, A. B. M. Aowlad Hossain.

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
