## [Decision Letter · Decision Letter 0]

27 Apr 2023

PONE-D-23-03551Investigation of the neural correlation with task performance and its effect on cognitive load level classificationPLOS ONE

Dear Dr. Khanam,

Thank you for submitting your manuscript to PLOS ONE. After careful consideration, we feel that it has merit but does not fully meet PLOS ONE’s publication criteria as it currently stands. Therefore, we invite you to submit a revised version of the manuscript that addresses the points raised during the review process.

We look forward to receiving your revised manuscript.

Kind regards,

Noman Naseer, PhD

Academic Editor

PLOS ONE

Journal Requirements:

2. Please note that PLOS ONE has specific guidelines on code sharing for submissions in which author-generated code underpins the findings in the manuscript. In these cases, all author-generated code must be made available without restrictions upon publication of the work. 

Please review our guidelines at https://journals.plos.org/plosone/s/materials-and-software-sharing#loc-sharing-code and ensure that your code is shared in a way that follows best practice and facilitates reproducibility and reuse.

**Additional Editor Comments:**

Reviewers see potential in this study. However, they have suggested some revisons. I encourage authors to revise upon those revisions and submit again.

Reviewers' comments:

Reviewer's Responses to Questions

**Comments to the Author**

1. Is the manuscript technically sound, and do the data support the conclusions?

Reviewer #1: Yes

Reviewer #2: Yes

2. Has the statistical analysis been performed appropriately and rigorously? 

Reviewer #1: Yes

Reviewer #2: No

3. Have the authors made all data underlying the findings in their manuscript fully available?

Reviewer #1: Yes

Reviewer #2: Yes

4. Is the manuscript presented in an intelligible fashion and written in standard English?

Reviewer #1: Yes

Reviewer #2: Yes

5. Review Comments to the Author

Reviewer #1: The authors investigate neural correlation with task performance and its effect on cognitive load level classification.

Though the paper is well structured and well written, I have these comments:

Authors mentioned , cognitive states (rest and task) from EEG signals are classified 97 considering a subject-independent approach. I barely see any results related to subject independent approach. Did authors used leave one person out cross validation?

Please check the images, most of them seems blur such as Fig 1, 2 and 4.

Among the participants, 47 were women and 19 were men. Did authors separately trained on women and tested on men or trained on women and tested on women? If yes, did authors noticed any difference?

In general, the data is more women dominant, did it effect the performance?

At the end, 36 the ultimate sample size was finalized for 36 (thirty-six) participants. Can we train a generalized person independent system based on 30+ participants?

Why SVM why not kNNs or decision trees or random forests?

What kernel authors used or tried? RBF? polynomial?

Reviewer #2: Overall, the paper has potential, but it needs to address the issues mentioned below to improve its quality and credibility. The authors should take these comments into consideration and make the necessary revisions to enhance the overall quality of the paper.

Table 1: The number of bad and good participants are counted incorrectly. The authors should correct the error in the table to ensure the accuracy of the data presented.

Image quality: The paper mentions that the image quality in Figure 4 is not good, and it is blurry. The authors should attach a clear picture in Figure 4 to improve the visual presentation of the results.

Proposed method: The paper mentions that the proposed method involves feature extraction and classification. However, it is unclear how the empirical mode decomposition method was used for feature extraction. The authors should provide more details on the method used to ensure that the readers can understand and replicate the study.

Discussion on findings: The paper should include a more detailed discussion on the findings to validate and compare the findings. This would provide a better understanding of the results and their significance. The authors should discuss the implications of the results, the limitations of the study, and the areas for future research.

It has been recommended that citations from the following papers be added to the paper to increase its credibility.

Asgher, U., Khan, M. J., Asif Nizami, M. H., Khalil, K., Ahmad, R., Ayaz, Y., & Naseer, N. (2021). Motor training using mental workload (MWL) with an assistive soft exoskeleton system: a functional near-infrared spectroscopy (fNIRS) study for brain–machine interface (BMI). Frontiers in Neurorobotics, 15, 605751.

Asgher, U., Khalil, K., Khan, M. J., Ahmad, R., Butt, S. I., Ayaz, Y., ... & Nazir, S. (2020). Enhanced accuracy for multiclass mental workload detection using long short-term memory for brain–computer interface. Frontiers in neuroscience, 14, 584.

Asgher, U., Ahmad, R., Naseer, N., Ayaz, Y., Khan, M. J., & Amjad, M. K. (2019). Assessment and classification of mental workload in the prefrontal cortex (PFC) using fixed-value modified beer-lambert law. IEEE Access, 7, 143250-143262.

6. PLOS authors have the option to publish the peer review history of their article (what does this mean?). If published, this will include your full peer review and any attached files.

Reviewer #1: **Yes: **Farzan Majeed Noori

Reviewer #2: No

---

## [Author Response · Author response to Decision Letter 0]

28 May 2023

Response to Reviewers

Ref : PONE-D-23-03551

Article Title : Investigation of the neural correlation with task performance and its effect on cognitive load level classification

Authors : Farzana Khanam, Mohiuddin Ahmad, and A.B.M. Aowlad Hossain

Additional Editor Comments:

Reviewers see potential in this study. However, they have suggested some revisions. I encourage authors to revise upon those revisions and submit again.

Response to Editor’s Comments:

Respected Editor, 

Thanks for your encouragement. We are happy that the reviewers see potential in this study. We appreciate constructive comments on our study as a way to improve. Therefore, we have minutely addressed all the comments of the respected anonymous reviewers and revised the manuscript accordingly. 

Sincere Regards,

Authors

 

Comments to the Author

Reviewer #1

The authors investigate neural correlation with task performance and its effect on cognitive load level classification. Though the paper is well structured and well written, I have these comments:

Comment #1: Authors mentioned, cognitive states (rest and task) from EEG signals are classified considering a subject-independent approach. I barely see any results related to subject independent approach. Did authors used leave one person out cross validation?

Response:

Thanks a lot for your appreciation, comments, and suggestions to improve the manuscript. There were 36 subjects performing mental arithmetic task. If the task were modeled and classified from subject-wise data, we could call it subject-depended classification results. Since, we consider the tasks independent to any subject rather only two tasks are considered rest and mental arithmetic, the result presented here is subject independent. To ensure the subject independency during training and evaluation of the classifier, we have used blind test dataset which was completely separated from the training and validation data.

Comment #2: Please check the images, most of them seems blur such as Fig 1, 2 and 4.

Response:

The uploaded image that accompanies the manuscript was of high quality, with a resolution of 300 dots per inch (dpi). Therefore, it should not have presented any issues regarding its clarity or resolution. However, it is possible that the image's quality was inadvertently reduced when it was converted into a PDF format for submission on the website.

Nevertheless, we have carefully reviewed the image and made efforts to improve its quality to ensure that it meets the highest standards. We have included the revised version of the image for your reference, and we are confident that it will provide the necessary visual support for the manuscript.

Fig 1. The brief depiction of performed activity schedule and storage time span during EEG recordings.

Fig 2. The complete experimental procedure at a glance.

Fig 4. The flowchart of extracting IMF features through a basic EMD algorithm.

Comment #3: Among the participants, 47 were women and 19 were men. Did authors separately trained on women and tested on men or trained on women and tested on women? If yes, did authors noticed any difference? In general, the data is more women dominant, did it effect the performance?

Response:

Initially the number of subjects was 66 (47 women and 19 men). However, after preprocessing, there were several data which had noise issue that’s why 30 data were excluded from the dataset by the dataset preparing and uploading team. The dataset is available contains of 36 subjects only (27 women and 9 men). Since our main focus is to investigate neural correlation with subject’s task performance, gender issue is quiet irrelevant with our research question of this study. Whether the subject is man or women, data with his/her ‘task experience’ is important for this research. Therefore, we have collected data of brain EEGs for cognitive load classification with ‘task experience’ scenarios and done experimental analysis in gender-independent way. Thank you for you insightful observation that the data is mostly from women subjects which could be future research scope for gender based analysis of cognitive loads. However, a bigger dataset with appropriate protocol will be necessary for such analysis to avoid any sensitive/conflicting gender issues. 

Comment #4: At the end, 36 the ultimate sample size was finalized for 36 (thirty-six) participants. Can we train a generalized person independent system based on 30+ participants?

Response:

Yes, after a rigorous checking, data of 36 subjects from 66 subjects has been finally used in this study. We agree that size of the dataset has impact on classifier learning. However, we have chosen this dataset as it is related to ‘task experience’ scenarios. Consequently, we have used conventional machine learning classifier (i.e. SVM) which exhibits proven performance for classification task with smaller dataset. We observed that, the experimental results of this subject independent system are found satisfactory which are given in Table 1. Availability of bigger data considering our research question is a constraint right now. However, we think the findings of this study might be useful for the future researches with bigger dataset and deep learning based classifier.

Comment #5: Why SVM why not kNNs or decision trees or random forests?

What kernel authors used or tried? RBF? polynomial?

Response:

For classification purposes several classifiers are initially deployed such as k-Nearest Neighbour, Naive Bayes, Decision Tree, SVM, and Linear Discriminant Analysis. However, SVM is used for the rest analysis of this work, as initially, SVM showed the highest result with polynomial of order 4. This explanation has been added into the manuscript and clarified with the results of other classifiers which are given in Table 1.

Reviewer #2

Overall, the paper has potential, but it needs to address the issues mentioned below to improve its quality and credibility. The authors should take these comments into consideration and make the necessary revisions to enhance the overall quality of the paper.

Comment #1 

Table 1: The number of bad and good participants are counted incorrectly. The authors should correct the error in the table to ensure the accuracy of the data presented.

Response:

We would like to express our sincere gratitude to the anonymous reviewer who brought this issue to our attention. Thanks to their diligent review and feedback, we were able to identify and correct the errors in the original manuscript. We have carefully addressed each of the reviewer's concerns and made the necessary revisions to ensure that the manuscript meets the highest standards of quality.

Once again, we extend our heartfelt appreciation to the reviewer for their invaluable contributions to this work. Their feedback has been instrumental in improving the accuracy and rigor of our research, and we are deeply grateful for their time and effort.

Comment #2 

Image quality: The paper mentions that the image quality in Figure 4 is not good, and it is blurry. The authors should attach a clear picture in Figure 4 to improve the visual presentation of the results.

Response:

We would like to draw your attention to the image that was uploaded along with the manuscript. It is important to note that the image was originally of very high quality, with a resolution of 300 dots per inch (dpi). This resolution should have ensured that the image was clear and detailed, with no issues related to its quality.

However, we suspect that the submission website may have inadvertently reduced the image's quality during the process of converting it into a PDF format. This is a common issue that can occur when images are converted from one format to another, and it can result in a loss of detail or clarity.

To address this concern, we have included the original version of the image for your reference. We hope that this will allow you to compare the image's original quality with the version that appears in the manuscript.

Fig 4. The flowchart of extracting IMF features through a basic EMD algorithm.

Comment #3 

Proposed method: The paper mentions that the proposed method involves feature extraction and classification. However, it is unclear how the empirical mode decomposition method was used for feature extraction. The authors should provide more details on the method used to ensure that the readers can understand and replicate the study.

Response:

Authors would like to clarify that the EEG signals are decomposed by EMD upto 6 levels as preprocessing of the raw EEG before features extraction. Since each levels are considered as IMF’s, 5 IMF’s are taken into consideration from where the features were extracted. From each IMF, the following features were extracted: mean, standard deviation (SD), skewness, kurtosis, root mean square (RMS), and the ratio of the absolute mean values of adjacent sub-bands. 

However, according to the reviewer’s guideline, the explanation is clearly presented in the revised paper. 

Comment #4: 

Discussion on findings: The paper should include a more detailed discussion on the findings to validate and compare the findings. This would provide a better understanding of the results and their significance. The authors should discuss the implications of the results, the limitations of the study, and the areas for future research.

Response:

We wish to express our sincere gratitude to the reviewer for their insightful comments and valuable contributions to our paper. We appreciate their effort in taking the time to engage with our work in such a thoughtful and constructive manner. These comments are addressed in the revised manuscript accordingly.

Comment #5:

It has been recommended that citations from the following papers be added to the paper to increase its credibility.

Asgher, U., Khan, M. J., Asif Nizami, M. H., Khalil, K., Ahmad, R., Ayaz, Y., & Naseer, N. (2021). Motor training using mental workload (MWL) with an assistive soft exoskeleton system: a functional near-infrared spectroscopy (fNIRS) study for brain–machine interface (BMI). Frontiers in Neurorobotics, 15, 605751.

Asgher, U., Khalil, K., Khan, M. J., Ahmad, R., Butt, S. I., Ayaz, Y., ... & Nazir, S. (2020). Enhanced accuracy for multiclass mental workload detection using long short-term memory for brain–computer interface. Frontiers in neuroscience, 14, 584.

Asgher, U., Ahmad, R., Naseer, N., Ayaz, Y., Khan, M. J., & Amjad, M. K. (2019). Assessment and classification of mental workload in the prefrontal cortex (PFC) using fixed-value modified beer-lambert law. IEEE Access, 7, 143250-143262.

Response:

These recent papers are cited accordingly as the reference number [7], [8], and [42]. 

Thanking to the reviewers again.

Authors

---

## [Decision Letter · Decision Letter 1]

24 Jul 2023

PONE-D-23-03551R1Investigation of the neural correlation with task performance and its effect on cognitive load level classificationPLOS ONE

Dear Dr. Khanam,

Thank you for submitting your manuscript to PLOS ONE. After careful consideration, we feel that it has merit but does not fully meet PLOS ONE’s publication criteria as it currently stands. Therefore, we invite you to submit a revised version of the manuscript that addresses the points raised during the review process.

The manuscript has addressed the reviewers' comments and feedback. However it is emphasized that figures (pictures) submitted with the manuscript are not of proper quality and resolution. Please improve the Picture quality and resubmit in the final version.

We look forward to receiving your revised manuscript.

Kind regards,

Umer Asgher, PhD

Academic Editor

PLOS ONE

Journal Requirements:

Additional Editor Comments (if provided):

The manuscript has addressed the reviewers' comments and feedback. However it is emphasized that figures (pictures) submitted with the manuscript are not of proper quality and resolution. Please improve the Picture quality and resubmit in the final version.

Reviewers' comments:

Reviewer's Responses to Questions

**Comments to the Author**

1. If the authors have adequately addressed your comments raised in a previous round of review and you feel that this manuscript is now acceptable for publication, you may indicate that here to bypass the “Comments to the Author” section, enter your conflict of interest statement in the “Confidential to Editor” section, and submit your "Accept" recommendation.

Reviewer #1: All comments have been addressed

Reviewer #2: All comments have been addressed

2. Is the manuscript technically sound, and do the data support the conclusions?

Reviewer #1: Yes

Reviewer #2: Yes

3. Has the statistical analysis been performed appropriately and rigorously? 

Reviewer #1: Yes

Reviewer #2: N/A

4. Have the authors made all data underlying the findings in their manuscript fully available?

Reviewer #1: Yes

Reviewer #2: Yes

5. Is the manuscript presented in an intelligible fashion and written in standard English?

Reviewer #1: Yes

Reviewer #2: Yes

6. Review Comments to the Author

Reviewer #1: The authors have revised the paper and addressed all the concerns. The manuscript has been improved, and I am happy to accept it in current form

Reviewer #2: I would like to extend my appreciation to you for resubmitting the manuscript with the improved version. I have carefully reviewed the revised manuscript, and I am pleased to inform you that all of the suggested comments that I provided have been diligently considered and incorporated into the resubmission.

I commend your commitment to addressing the concerns and suggestions raised during the initial review process. Your responsiveness and attention to detail in implementing the necessary revisions have significantly enhanced the quality and clarity of your work.

Thank you again for your hard work and dedication. I look forward to seeing your research published.

7. PLOS authors have the option to publish the peer review history of their article (what does this mean?). If published, this will include your full peer review and any attached files.

Reviewer #1: **Yes: **Farzan Majeed Noori

Reviewer #2: No

---

## [Author Response · Author response to Decision Letter 1]

28 Aug 2023

Dear Editor,

Due to the concern of image quality, we revised the image quality and improved them accordingly. We hope the revised edition of the images will fulfill the required quality. The figure may found degraded in the PDF due to compression, but the uploaded figures are of 300dpi. For any confusion, plz check the uploaded images in the submission portal. 

Besides, both the previous corrections in the manuscript with track changes and unmarked version of our revised paper without tracked changes are uploaded accordingly. 

Thank you again for your effort and contribution to improve our manuscript. We are happy to provide more information to publish this article. 

Authors

---

## [Editor Report · Decision Letter 2]

1 Sep 2023

Investigation of the neural correlation with task performance and its effect on cognitive load level classification

PONE-D-23-03551R2

Dear Dr. Khanam,

We’re pleased to inform you that your manuscript has been judged scientifically suitable for publication and will be formally accepted for publication once it meets all outstanding technical requirements.

Kind regards,

Umer Asgher, PhD

Academic Editor

PLOS ONE

---

## [Editor Report · Acceptance letter]

8 Sep 2023

PONE-D-23-03551R2 

Investigation of the neural correlation with task performance and its effect on cognitive load level classification 

Dear Dr. Khanam:

I'm pleased to inform you that your manuscript has been deemed suitable for publication in PLOS ONE. Congratulations! Your manuscript is now with our production department. 

Kind regards, 

on behalf of

Dr. Umer Asgher 

Academic Editor

PLOS ONE